# Wernicke Encephalopathy Mimicking MELAS

**DOI:** 10.3390/medicina58050660

**Published:** 2022-05-13

**Authors:** Josef Finsterer

**Affiliations:** Neurology & Neurophysiology Center, Postfach 20, 1180 Vienna, Austria; fifigs1@yahoo.de; Tel./Fax: +43-1-5861075

**Keywords:** thiamine deficiency, Wernicke encephalopathy, stroke-like lesion, stroke-like episode, lactic acidosis, spectroscopy

## Abstract

Objectives: a stroke-like lesion, the morphological equivalent of a stroke-like episode and the hallmark of mitochondrial encephalopathy, lactic acidosis, and stroke-like episodes (MELAS) syndrome, have not been reported as manifestations of thiamine deficiency. Case report: a 62-year-old man with a history of chronic alcoholism was admitted after a series of epileptic seizures. Upon waking up from the coma, he presented with disorientation, confusion, confabulation, psychomotor agitation, aggressiveness, right hemianopsia, aphasia, and right hemineglect over weeks. Electroencephalography showed a questionable focal status epilepticus over the left hemisphere, responsive to lorazepam and oxcarbazepine. Follow-up electroencephalographies no longer recorded epileptiform discharges. Cerebral magnetic resonance imaging (MRI) revealed T2-/diffusion weighted imaging (DWI) hyperintensity in the left occipito-temporal region that was not congruent to a vascular territory which persisted for at least nine weeks. Since a lactate-peak could be seen in this lesion by magnetic resonance-spectroscopy, this was interpreted as a stroke-like lesion. Since thiamine was reduced, the stroke-like lesion was attributed to thiamine deficiency after the exclusion of differential diseases, including MELAS and status epilepticus. The patient’s behavioural and cognitive dysfunctions largely resolved upon vitamin-B1 substitution. Conclusions: the case suggests that thiamine deficiency presumably causes mitochondrial dysfunction with cerebrospinal fluid lactic acidosis and a stroke-like lesion mimicking MELAS syndrome. It should be further studied whether nutritional deficits, such as thiamine deficiency, could give rise to secondary stroke-like lesions.

## 1. Introduction

Wernicke encephalopathy (WE) is due to thiamine deficiency and clinically characterised by ataxia, ophthalmoparesis, and confusion in <25% of cases [1,2]. Additional features can be dysarthria, neuropathy, hypotonia, and non-specific abnormalities, such as loss of appetite, headache, fatigue, impaired concentration, irritability, abdominal discomfort, nausea, and vomiting [3,4]. Cerebral magnetic resonance imaging (MRI) typically shows symmetric T2-hyperintense lesions in the thalami and the periaqueductal grey matter [1]. A stroke-like lesion, the morphological equivalent of a stroke-like episode and the hallmark of mitochondrial encephalopathy, lactic acidosis, and stroke-like episodes (MELAS), has not been reported as manifestations of malnutrition, resulting in thiamin deficiency.

## 2. Case Report

The patient is a 62-year-old Caucasian, shelterless male, height 180 cm, who was admitted after a series of five focal epileptic seizures with secondary generalization in four of them, presumably triggered by a non-specific infection. All seizures were witnessed and lasted <5 min each. His history was positive for severe chronic alcoholism over the years, retinal detachment at age 41 years, duodenal ulcers, and erysipelas. Chronic alcoholism resulted in liver cirrhosis, hepato-splenomegaly with consecutive thrombocytopenia, esophageal varicositas, chronic venous insufficiency, leg edema, and malnutrition. 

A clinical neurologic exam on admission revealed a coma, conjugated gaze palsy to the right, mild right-sided hemiparesis, and positive pyramidal signs. After awakening, the patient presented with disorientation, confusion, confabulation, psychomotor restlessness, agitation, aggressiveness, hemianopsia to the right, aphasia, and hemineglect to the right. Psychomotor excitement was accompanied by aggressive tendencies and the unbreakable misjudgement of his situation and environment. MRI of the cerebrum revealed global atrophy, and a T2-/diffusion weighted imaging (DWI) hyperintense lesion in the left occipito-temporal area not confined to a vascular territory, which was slightly hypointense on apparent diffusion coefficient (ADC) being interpreted as a stroke-like lesion. This lesion persisted over the weeks (Figure 1). There was also a small T2-hyperinetense lesion in the left thalamus and the right parietal cortex. Magnetic resonance (MR)-spectroscopy, carried out nine weeks after admission, revealed an intra-lesional lactate-peak, but normal NAA/Cr-ratio. Electroencephalography two days after admission showed continuous spikes and sharp waves over the entire left hemisphere, with phase reversal T5 being interpreted as focal status epilepticus. The anti-seizure drug therapy included lorazepam and oxcarbazepine. Transthoracic echocardiography was non-informative. 

Blood tests revealed a negative alcohol level, mild macrocytic anaemia, leucocytosis, thrombopenia, elevated creatine-kinase, and normal NH3 but reduced vitamin-B12 and thiamine levels (Table 1). The virus panel was non-informative. Vitamin substitution, including for vitamin B1, B2, B6, and B12, was started. Due to the patient’s behavioural and cognitive alterations, neuro-psychological testing was not feasible during hospitalization. 

Since the psychiatric abnormalities were not sufficiently manageable on the neurological department, the patient was transferred to the psychiatric department and scheduled for a nursery home. 

At the last follow-up 16 years after the event, the patient had made a remarkable recovery and was living in a nursery home with a modified Rankin Scale (mRS) of 2.

## 3. Discussion

The patient is interesting for a stroke-like lesion presumably triggered by thiamine deficiency, which was previously unreported. He is also noteworthy for WE without the typical symmetric MRI features, but with an asymmetric thalamic lesion, a right cortical lesion, and a left stroke-like lesion. Stroke-like lesions are characterised by T2-, FLAIR-, DWI-hyperintensity, hyperperfusion on perfusion-weighted imaging, hypointensity on OEF-MRI, hypometabolism on FDG-PET, and a lactate-peak on MRS. ADC can be highly variable. Stroke-like lesions are typically not confined to a vascular territory. In this case, stroke-like lesions were primarily attributed to thiamine deficiency, as the patient did not fulfil the Japanese or Hirano criteria for diagnosing MELAS [5,6], for which stroke-like lesions are the hallmark [7].

Arguments against MELAS are that the family history was negative for a mitochondrial disorder, the age of 62 years, the height of 180 cm, absence of dysmorphisms, diabetes, hypoacusis, cardiomyopathy, and myopathy, normal renal function, and the normalisation of serum creatine-kinase, and the cerebrospinal fluid lactate after discontinuation of seizures (Table 1). A further argument against MELAS is that the psychiatric abnormalities resolved with vitamin B1 substitution, and that no phenotypic features of MELAS developed during a follow-up of 16 years. 

Arguments in favour of MELAS are the retinal detachment, the stroke-like episode with aphasia, hemianopia, hemineglect, and hemiparesis, the corresponding stroke-like lesion, and the lactate-peak, which persisted for at least nine weeks after admission. However, retinal detachment can be due to several other causes. Though lactate-peaks most frequently occur in mitochondrial disorders, they have been repeatedly reported in patients with thiamine deficiency [8,9]. Though stroke-like lesions are regarded as pathognomonic for MELAS, they occur in other primary mitochondrial disorders as well. Since thiamine deficiency causes a secondary mitochondrial disorder, the stroke-like lesion in the index patient was attributed to thiamine deficiency. 

A status epilepticus was excluded as the cause of the MRI lesions, since the patient never had a status epilepticus clinically, and as a status, epilepticus was never recorded with certainty on EEG. Only in the first EEG, two days after admission, was a questionable focal status or periodic lateralized epileptiform discharges (PLEDS) documented. In none of the three following EEGs was a status epilepticus visible. A further argument against status epilepticus or seizure activity as the cause of the MRI lesions is that these lesions persisted for at least nine weeks until the last MRI after admission. Though MRI lesions resembling those of the index patient have been previously reported in patients with a status epilepticus, they do not persist over such a long period of time as in the index case. Though the index patient had a left thalamic lesion, it was definitively not a “pulvinar sign”, as occasionally reported in patients with a status epilepticus.

Alcohol encephalopathy was excluded, as the alcohol serum level was negative. Hepatic encephalopathy was excluded from the normal NH3. Viral encephalitis was excluded from the negative virus panel and absence of pleocytosis. Accordingly, the most plausible explanation for cerebral abnormalities is WE. Arguments in favour of WE are the low level of thiamine and previous reports, showing that WE and thiamine deficiency can go along with lactic acidosis [10], and even with a lactate-peak on magnetic resonance spectroscopy [9] and with a low NAA/Cr-peak [9]. The latter was not documented in the index patient. Lactate-peaks and low NAA/Cr-peak normalise after thiamine substitution [8]. The right parietal lesion can be attributed to thiamine deficiency as well, as cortical lesions with seizures have been reported as the initial manifestations of WE [9]. The left thalamic lesion is also in line with the imaging findings in thiamine deficiency (pulvinar sign, hockey stick sign).

The pathophysiological mechanism responsible for thiamine deficiency-induced mitochondrial dysfunction is poorly understood. There is some evidence that thiamine deficiency can induce oxidative stress and thereby impair mitochondrial functions. There is also some evidence that thiamine deficiency results in the reduced synthesis of thiamine pyrophosphate, an active cofactor of the pyruvate dehydrogenase E1α (PDHE1α) [10]. Low PDHE1α activates the pyruvate dehydrogenase kinase, which in turn results in the phosphorylation of PDHE1α. Phosphorylated PDHE1α blocks oxidative phosphorylation and thus ATP production. 

Limitations of the study are that no MRI after the resolution of the DWI hyperintensities is available, that no neuropsychological investigation could have been performed, and that no genetic testing had been carried out.

In conclusion, the case suggests that thiamine deficiency can be associated with cerebrospinal fluid lactic acidosis, and that stroke-like lesions may not only occur in genetic mitochondrial disorders, but also as a result of secondary deficits, such as thiamine deficiency, mimicking MELAS. Further studies are required to assess whether nutritional deficits, such as thiamine deficiency, could give rise to secondary stroke-like lesions.

## Figures and Tables

**Figure 1 medicina-58-00660-f001:**
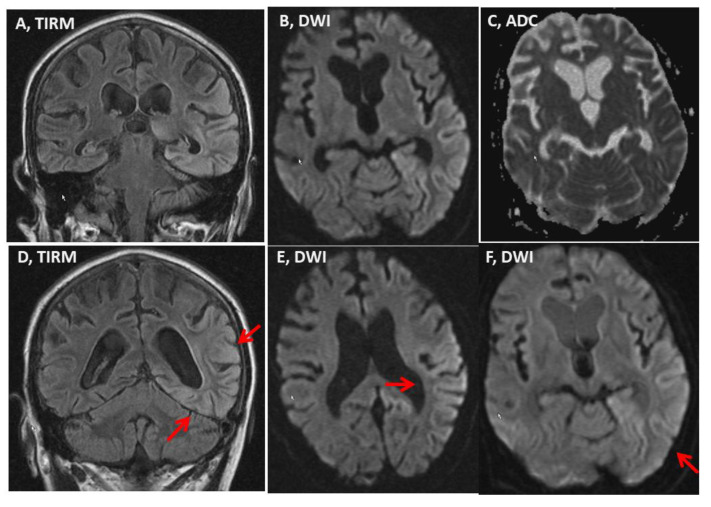
Magnetic resonance imaging (MRI) of the brain carried out on hospital day 13 showing a T2 and diffusion weighted imaging DWI hyperintense lesion in the left occipito-temporal region and the median left thalamus (panels **A**,**B**,**D**,**E**,**F**), which were slightly hypointense on apparent diffusion coefficient (ADC) (panel **C**). The MRI findings were unchanged compared to those on hospital day 1 and hospital day 3.

**Table 1 medicina-58-00660-t001:** Relevant blood tests obtained during hospitalisation.

Parameter	Reference Limits	hd1	hd3	hd7	hd13	hd20
Leucocytes	4.0–9.0/nl	**12.9**	6.7	6.7	7.5	**9.8**
Erythrocytes	4.25.5/pl	**3.25**	**3.35**	**3.17**	**3.13**	**3.17**
MCV	80–98 fl	**99.1**	**98.7**	**99.1**	**99.7**	**100.1**
Thrombocytes	150–450/nl	**40**	**44**	**103**	175	218
Creatine-kinase	<170 U/l	**2485**	**1966**	**683**	110	nd
GGT	<54 U/l	**1528**	nd	nd	648	553
NH3	10–48 μmol/l	nd	42	nd	nd	nd
Alcohol	<50 mg/dl	<50	nd	nd	nd	nd
Vitamin B12	145–569 pmol/L	nd	nd	nd	nd	**137**
Vitamin B1	33.1–60.7 μg/L	nd	nd	nd	nd	**17.2**
CSF cells	<13/3	4/3	nd	12/3	7/3	nd
CSF lactate	0–2.1 mmol/l	**4.4**	nd	**4.59**	1.58	nd

CSF: cerebrospinal fluid, GGT. Gamma-glutamyl transferase, hd: hospital day, MCV: mean cell volume, nd: not determined; bold cahracters: these are the abnornall ones.

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
