# Peer review of "Wernicke Encephalopathy Mimicking MELAS"

_medicina, 2022, doi:10.3390/medicina58050660_

Round 1

Reviewer 1 Report

Title: Wernicke encephalopathy mimicking MELAS

Manuscript ID: Medicina- 1650875

This is an interesting manuscript about a case of Wernicke encephalopathy c MRS study.

In spite of theses attractive results, some careful considerations should be made.

Major point
1. MELAS syndrome is a rare disorder that begins in childhood related to Mitochondrial disorder.

This case was not related to MELAS syndrome at all, because this patient does not have genetic confirmation.

And we can explain this case as a status epilepticus related to Wernicke encephalopathy or chronic alcoholics.

I think this is simple case about SE. Why did you mention about MELAS?

  1. MELAS syndrome usually begin in childhood between 2 and 15 years of age.

Because it is a syndrome, many symptoms such as lactic acidosis, stroke like episodes and seizure can be observed in other disease.

But we do not mention it as MELAS syndrome.

  1. In this case, MRI findings with T2 high, low ADC can be related to SE.

I cannot catch the novel finding in this case.

If you have another point of view, please clarify it.

  1. In abstract and manuscript,

A lot of abbreviations (SLL, MID, SLE, RL, hd) seems to be unnecessary.

5 In Figure,

Appropriate explanations about 6 panels should be needed.

It would be better to present those panels as Figure 1A,B,C,D,E, and F.

Minor point

  1. In main body, line 50,

Hyperintemse lesion à hyperintense lesion

Author Response

This is an interesting manuscript about a case of Wernicke encephalopathy c MRS study. In spite of theses attractive results, some careful considerations should be made.
Major point

MELAS syndrome is a rare disorder that begins in childhood related to Mitochondrial disorder.

We do not agree. MELAS can have an adult onset as well [e.g. Kamath et al. 2022].

This case was not related to MELAS syndrome at all, because this patient does not have genetic confirmation. And we can explain this case as a status epilepticus related to Wernicke encephalopathy or chronic alcoholics. I think this is simple case about SE. Why did you mention about MELAS?

Seizure-related cerebral lesions usually resolve within a few hours or days. The lesion in the index persisted during at least 9 weeks. Another argument against a seizure-related radiological phenomenon is that the patient had generalised seizures but a focal radiological lesion. Another argument against the seizure hypothesis is that MRS 9 weeks after admission showed a lactate peak in the absence of seizures. The differential diagnosis MELAS was discussed until discharge because thiamine deficiency causes secondary mitochondrial disorder and because stroke-like lesions are the hallmark of MELAS.      

MELAS syndrome usually begin in childhood between 2 and 15 years of age. Because it is a syndrome, many symptoms such as lactic acidosis, stroke like episodes and seizure can be observed in other disease. But we do not mention it as MELAS syndrome.

MELAS can have a late onset as well [Landis 2021, Kamath et al. 2022]. At onset MELAS can be mono-symptomatic. We do not agree that stroke-like episodes occur in diseases other than mitochondrial disorders. Metabolic stroke (i.e. stroke-like episodes) are typical for primary or secondary mitochondrial disorders). The combination of seizures, lactic acidosis, and a stroke-like episode, which persists over weeks, is pathognomonic for a mitochondrial disorder, including a secondary mitochondrial disorder due to thiamine deficiency. The lactate peak on MRS supports the suspicion of a SLL.     

In this case, MRI findings with T2 high, low ADC can be related to SE. I cannot catch the novel finding in this case. If you have another point of view, please clarify it.

The novelty is that thiamine deficiency can cause a secondary mitochondrial disorder and can manifest with a stroke-like episode. The strongest argument against a para epileptic phenomenon is that the cerebral stroke-like lesion persisted over weeks. 

In abstract and manuscript,

A lot of abbreviations (SLL, MID, SLE, RL, hd) seems to be unnecessary.

These abbreviations were removed

In figure, Appropriate explanations about 6 panels should be needed.

It would be better to present those panels as Figure 1A,B,C,D,E, and F.

The 6 panels are now labelled as suggested

In main body, line 50, Hyperintemse lesion à hyperintense lesion

Thank you. Was corrected

Reviewer 2 Report

Dear Dr. Josef Finsterer, I read and reviewed with great interest your article. I made some remarks which I hope to be fulfilled. 

Congratulations for your work.

My best regards.

Author Response

I read and reviewed with great interest the article by Dr. J. Finsterer. He described a case of Wernicke encephalopathy (WE) presenting with clinical signs mimicking a stroke-like episode (SLE). He also highlighted the peculiarity of the MRI findings, corresponding to a stroke-like lesion (SLL). Both these clinical and MRI elements are often associated with mitochondrial diseases, such as MELAS, and SLL has never been attributed to thiamine deficiency before. The rationale of the article implies that thiamine deficiency can lead to a secondary mitochondrial impairment causing neurological and neuroradiological findings that overlap primary mitochondrial diseases.

Thank you.

Overall, this case report proposes an original theory underlying the relationship between WE and mitochondrial impairment clinical manifestations; it is well-organized and the English language is clear, although there are minor spell mistakes and phrasing issues that I list as follows: - Line 27: …ataxia. ophthalmoparesis… → …ataxia, ophthalmoparesis…; - Line 45 and 85: hemianopia → hemianopsia; - Line 50: T2-hyperintemse → T2-hyperintense; - Line 96: lactate-peal → lactate-peak.

Thank you. Was corrected

I have also some major remarks regarding the diagnostic work-up and the discussion of the case report as follows:

  1. There are some other cases of SLE associated with WE in Literature 1-2, although it is the first time that a SLL is attributed to a thiamine deficiency. The correlation seems likely, although some MRI features could be attributed merely to a vascular stroke. Specifically, the left occipito-temporal lesion showed an hypointensity on ADC, which is also common for a recent ischemic stroke. Were any additional perfusion-based exams performed, like CT-perfusion or PW-MRI, in order to rule out a vascular etiology rather than a metabolic one? I would expect to find an increase of perfusion parameters in SLL3, unlike in a vascular stroke, which could be consistent with metabolic hypothesis.

1 Bhan A, Advani R, Kurz KD, Farbu E, Kurz MW. Wernicke's Encephalopathy Mimicking Acute Onset Stroke Diagnosed by CT Perfusion. Case Rep Neurol Med. 2014;2014:673230. doi: 10.1155/2014/673230.

2 Chang GY. Acute Wernicke's syndrome mimicking brainstem stroke. Eur Neurol. 2000;43(4):246-7. doi: 10.1159/000008168. PMID: 10828660.

3 Li Y, Xu W, Sun C, Lin J, Qu J, Cao J, Li H, Yang L. Reversible Dilation of Cerebral Macrovascular Changes in MELAS Episodes. Clin Neuroradiol. 2019 Jun;29(2):321-329. doi: 10.1007/s00062-018-0662-8.

Reviewing the 5 MRIs of our pat. carried out during the 3 months of hospitalisation, we could not find a perfusion study but we agree that stroke-like lesions are characterised by hyperperfusion, as reported by Li in their 14 MELAS patients. Whether the hyper-perfused thalamic lesions in Bhan’s case truly represent stroke-like lesion, remains speculative. No multimodal MRI of their patient was provided. Since the territory of the lesion in our patient was not confined to any vascular territory, an ischemic stroke is rather unlikely. ADC can be hypointense in typical stroke-like lesions on ADC [Xu et al. 2018]. The patient reported by Chang did not experience a typical SLL as it occurred in the brainstem. A definition of a SLL was provided in the first paragraph of the discussion.

  1. In addition to the previous annotation, since ischemic stroke could have been the cause of the patient’s focal symptoms, were any ancillary tests performed to exclude a potential ischemic etiology, besides the transthoracic echocardiography? For example, EKG prolonged monitoring or vascular assessment (supra-aortic trunks doppler ultrasonography, CT or MRI Angiography).

MR angiography was normal. CTA had not been carried out. Classical cardiovascular risk factors were absent. Standard ECG showed sinus rhythm. No Holter monitoring had been carried out. There was no arterial hypertension. Based on these findings and the MRI, ischemic stroke was excluded.

  1. In the author's opinion, which could be the nature of the T2-hyperintense lesions in the left thalamus and the right parietal cortex? What were their appearances on DWI/ADC? Could they be considered simultaneous to the left SLL, and then share the same etiology, or could they be attributed to a different cause, considering their localization, size, and MRI features?

Thalamic lesions are typical for thiamine deficiency (pulvinar sign) [Khan 2020]. They do not need to be symmetric. Cortical lesions with seizures have been reported as the initial manifestation of WE [Fu et al. 2017].  We agree that it cannot be excluded that the right parietal lesion represents an abortive stroke-like lesion. However, this lesion did not show dynamic changes over time.

  1. As pointed out by the author, the cause-and-effect relationship between the thiamine deficiency and the SLL is explained by thiamine deficiency leading to a mitochondrial impairment and thus mimicking a primary mitochondrial disease. I would like the author to develop this concept, briefly explaining the possible pathophysiology mechanisms involved, and citing some articles on support. I would recommend this one4.

4 Abdou E, Hazell AS. Thiamine deficiency: an update of pathophysiologic mechanisms and future therapeutic considerations. Neurochem Res. 2015 Feb;40(2):353-61. doi: 10.1007/s11064-014-1430-z.

A new paragraph was created to discuss the pathophysiological mechanisms how thiamine deficiency can impair mitochondrial functions. The explanations are in line with the mechanisms discussed in Abdou et al. 2015.

  1. The author provides arguments against and in favour of the differential diagnosis between MELAS and WE, using the Hirano and Yatsuga (or Japanese) criteria. Specifically, Yatsgua criteria require at least two A category and two B category criteria to be met5 . Since the patient presented either seizures and acute focal lesions on neuroimaging (A) or high CSF lactate levels (B), could a research of mitochondrial abnormalities on muscle biopsy have helped as additional criteria of the B category? If positive, could this have led to the diagnosis of MELAS?

5 Yatsuga S, Povalko N, Nishioka J, Katayama K, Kakimoto N, Matsuishi T, Kakuma T, Koga Y; Taro Matsuoka for MELAS Study Group in Japan. MELAS: a nationwide prospective cohort study of 96 patients in Japan. Biochim Biophys Acta. 2012 May;1820(5):619-24. doi: 10.1016/j.bbagen.2011.03.015.

Diagnosing MELAS requires documentation of a pathogenic mutation. Muscle biopsy is not sufficient to diagnose MELAS. Muscle biopsy can be helpful to detect the causative mtDNA variant, as mtDNA deletions are best detected in skeletal muscle.

In addition, some minor remarks: 1. Were the serum lactate levels dosed, besides on the CSF? 2. Regarding the patient follow-up: were MRI sequences repeated in the following years during recovery? If available, these data could support the benefit of thiamine substitution.

Serum lactate was determined on admission but was normal. Follow-up MRI was recommended but since the patient was institutionalised in a nursery home, forgotten to be accomplished. 

Reviewer 3 Report

The paper is concise and the MRI pictures are of acceptable quality showing gyral and basal ganglia diffusion restriction in the left temporo-occipital area which appears hypointense on ADC mapping.

The author interprets these MRI changes as typical of a mitochondrial disease and draws the conclusion that thiamine deficiency may trigger a secondary mitochondrial disorder with consecutive energy deficit.

The patient did, however, also present with a focal status epilepticus originating in the left temporo-occipital region. All MRI changes shown are also well-known consequences of sustained seizure activity, resulting in tissue hypoxia and eventually cortical laminar necrosis.
The differential diagnosis of gyriform restricted diffusivity and low density on ADC includes – apart from status epilepticus – hypoxic encephalopathy, viral infections and metabolic encephalopathies. In most mitochondrial-induced stroke-like episodes there is little or no change of the ADC signal or indeed even an increased ADC signal is seen with additional involvement of the subcortical white matter.

Further, intralesional lactate peaks and elevated CSF lactate are also observed after prolonged status epilepticus.

Whilst I agree that in this patient the MRI and CSF changes indicate a breakdown of local cellular energy metabolism, this breakdown is a direct consequence of the status epilepticus and not specific for a mitochondrial disorder. Any metabolic disorder leading to status epilepticus may give rise to similar changes.

I would therefore recommend the author not to put too much emphasis on the similarity to mitochondrial disorders, but rather on the status epilepticus, and then resubmit the paper.

Author Response

The paper is concise and the MRI pictures are of acceptable quality showing gyral and basal ganglia diffusion restriction in the left temporo-occipital area which appears hypointense on ADC mapping.

Thank you.

The author interprets these MRI changes as typical of a mitochondrial disease and draws the conclusion that thiamine deficiency may trigger a secondary mitochondrial disorder with consecutive energy deficit.

The patient did, however, also present with a focal status epilepticus originating in the left temporo-occipital region. All MRI changes shown are also well-known consequences of sustained seizure activity, resulting in tissue hypoxia and eventually cortical laminar necrosis.

Seizures stopped upon application of anti-seizure drugs. Follow-up EEGs did no longer record epileptiform discharges. MRI lesions following seizure activity are usually transient and do not persist for at least 9 weeks as in our case.

The differential diagnosis of gyriform restricted diffusivity and low density on ADC includes – apart from status epilepticus – hypoxic encephalopathy, viral infections and metabolic encephalopathies. In most mitochondrial-induced stroke-like episodes there is little or no change of the ADC signal or indeed even an increased ADC signal is seen with additional involvement of the subcortical white matter.

We agree that ADC can be hyper- hypo- or iso-intense in SLLs. Therefore, ADC maps do not contribute much to the characterisation of SLLs. Modalities like DWI, PWI, and OEF are more helpful in this respect. The patient had undergone CSF investigations without indication for meningitis/encephalitis.

Further, intralesional lactate peaks and elevated CSF lactate are also observed after prolonged status epilepticus.

We agree, but the lactate peak was still present nine weeks after admission. Since there were no seizures since admission and since EEGs did not record epileptiform discharges, seizures are rather unlikely to be responsible for elevated cerebral lactate

Whilst I agree that in this patient the MRI and CSF changes indicate a breakdown of local cellular energy metabolism, this breakdown is a direct consequence of the status epilepticus and not specific for a mitochondrial disorder. Any metabolic disorder leading to status epilepticus may give rise to similar changes.

I would therefore recommend the author not to put too much emphasis on the similarity to mitochondrial disorders, but rather on the status epilepticus, and then resubmit the paper.

We do not agree. The left temporo-occipital lesion could be seen even 9 weeks after admission. Seizures were seen only on admission and did not recur during the entire hospitalisation of three months.   

Reviewer 4 Report

The author presents an interesting and clearly written case of a stroke-like episode with a possible association with thiamine deficiency. The possibility of such a differential diagnosis would be a novelty and a valuable piece of clinical information, which also has theoretical relevance considering the role of thiamine in energy metabolism.

However, the case study raises many questions which should be clarified.

  1. medical history: had the patient psychomotor retardation of any degree primarily? Previous stroke-like episodes or epilepsy? Any information on family history?
  2. the central clinical hallmarks of WE should be described in more detail (or absence thereof!). The clinical presentation on admission is probably due to ongoing seizure activity. Which were the WE findings after awakening? Ophthalmoplegia? Conjugate gaze palsies? Vestibular disorder & ataxia? It would also be valuable to know of the clinical course of the central symptoms in more detail.
  3. It should be argued why the MRI (and MRS & CSF) findings are not explained by epileptic activity, as well as - and particularly - the absence of characteristic symmetric midline lesions (mammillary bodies, periventricular region) in areas known to be most sensitive to derangements of energy metabolism due to thiamine deficiency.
  4. It should be clarified when the thiamine level was determined at the acute-subacute stage (the table gives only day 20).
  5. Was the patient administered thiamine substitution? And at which stage? If thiamine deficiency was present throughout the acute stage, it would be expected that the symptoms attributable to B1 deficiency would be present and even accentuated especially after glucose administration.
  6. What was the mRS2 due to? Were there subsequent clinical controls or any neuropsychological examination at any stage (which would be potentially highly valuable accessory information considering the diagnosis of WE)?
  7. The case would be made considerably stronger by exclusion of genetic defects associated with primary mitochondrial disorder (e.g. POLG). Could genetic testing reveal a latent primary disposition with a propensity to SLL? Absence of genetic testing could be commented on.
  8. the conclusion as presented in the abstract and at the end of the discussion section is too strong. The data presented can not be taken as a demonstration that thiamine deficiency can cause a stroke-like episode on the basis of a secondary mitochondrial disorder (overall, it is debatable whether a single patient case could ever provide such evidence). At most, the case may serve to raise the question whether thiamine deficiency could predispose to SLE, at least in a subpopulation of patients (e.g. with some other predisposing factor), and thus prompt further study.

Author Response

The author presents an interesting and clearly written case of a stroke-like episode with a possible association with thiamine deficiency. The possibility of such a differential diagnosis would be a novelty and a valuable piece of clinical information, which also has theoretical relevance considering the role of thiamine in energy metabolism.

Thank you.

However, the case study raises many questions which should be clarified.

  1. medical history: had the patient psychomotor retardation of any degree primarily? Previous stroke-like episodes or epilepsy? Any information on family history?

There was no psychomotor retardation. As far as electronic records tell, there was no previous stroke-like episode. The family history was not provided in detail but according to available electronic records, the family history was negative for MELAS. .

  1. the central clinical hallmarks of WE should be described in more detail (or absence thereof!). The clinical presentation on admission is probably due to ongoing seizure activity. Which were the WE findings after awakening? Ophthalmoplegia? Conjugate gaze palsies? Vestibular disorder & ataxia? It would also be valuable to know of the clinical course of the central symptoms in more detail.

WE manifestations after awaking were confusion, disorientation, agitation, aggressiveness, and increased motor activity, as described in the report. Ophthalmoplegia was never recognised. No seizures were observed. EEGs were non-informative. Hemianopsia to the right, hemineglect, and aphasia persisted after awakening. There was psychomotor excitement with aggressive tendencies in unbreakable misjdugement of his situation and environment. There was no vestibular disorder but mild ataxia. This was added to the case description. 

  1. It should be argued why the MRI (and MRS & CSF) findings are not explained by epileptic activity, as well as - and particularly - the absence of characteristic symmetric midline lesions (mammillary bodies, periventricular region) in areas known to be most sensitive to derangements of energy metabolism due to thiamine deficiency.

MRI features persisted over weeks. The lactate peak was documented 9 weeks after admission. Cortical and thalamic lesions are well described in WE and not necessarily need to be symmetric [Welch et al. 1996].

  1. It should be clarified when the thiamine level was determined at the acute-subacute stage (the table gives only day 20).

Was determined only once during hospitalisation.

  1. Was the patient administered thiamine substitution? And at which stage? If thiamine deficiency was present throughout the acute stage, it would be expected that the symptoms attributable to B1 deficiency would be present and even accentuated especially after glucose administration.

The patient received vitamin substitution since admission but B1 intravenously not earlier than after documentation of B1 deficiency. Aggravation of WE symptoms was not observed after glucose rich meals. Glucose was not administered intravenously.   

  1. What was the mRS2 due to? Were there subsequent clinical controls or any neuropsychological examination at any stage (which would be potentially highly valuable accessory information considering the diagnosis of WE)?

The patient was institutionalised in a nursery home after discharge. There were no clinical controls. At a telephone follow-up 16 years after discharge, the patient was described as mobile with a walker and with mild disability, assessed as mRS 2. According to the available records he did not undergo a detailed neuropsychological exam.

  1. The case would be made considerably stronger by exclusion of genetic defects associated with primary mitochondrial disorder (e.g. POLG). Could genetic testing reveal a latent primary disposition with a propensity to SLL? Absence of genetic testing could be commented on.

We agree. However, genetic testing was not feasible during hospitalisation due to legal reasons. It was not carried out either in the nursery home.

  1. the conclusion as presented in the abstract and at the end of the discussion section is too strong. The data presented can not be taken as a demonstration that thiamine deficiency can cause a stroke-like episode on the basis of a secondary mitochondrial disorder (overall, it is debatable whether a single patient case could ever provide such evidence). At most, the case may serve to raise the question whether thiamine deficiency could predispose to SLE, at least in a subpopulation of patients (e.g. with some other predisposing factor), and thus prompt further study.

The conclusions were modified accordingly.

Round 2

Reviewer 1 Report

N/A 

Author Response

Reviewer 3

As stated in my initial comments, sustained status epilepticus which occurred in this patient, has to be considered and discussed. I could not see any mention of this important differential in the revised version of the manuscript

A new paragraph was created in the discussion. There the differential “status epilepticus” (SE) is now discussed in more detail.

Status epilepticus is well-known to cause similar changes as shown in this patient, and it is indeed the most likely reason for the MRI abnormalities. In particular, a pulvinar sign is often also seen as a consequence of status epilepticus.

Our patient did not present with a “pulvinar sign. There was a thalamic hyperintensity but this was not located in the pulvinar of the thalamus.  Second, when searching pubmed for „status epilepticus“ and “pulvinar sign” there is no appropriate hit. Therefore, we do not agree that the pulvinar sign is often seen in SE.

The author argues that the persistence of the MRI changes over 9 weeks excludes status epilepticus as a possible cause. However, the drop in the ADC values reflects permanent cortical damage (cortical laminar necrosis) which can also be a consequence of status epilepticus, independent of its etiology. Thus, persistence of the MRI changes per se does not exclude SE as the cause of the MRI abnormalities.

Cortical laminar necrosis is seen in prolonged SE. However, the patient never had a prolonged SE. Second, the last MRI, 9 weeks after admission, does not show the final outcome of these lesions. Whether the SLL truly ended up as laminar cortical necrosis could be assessed only after resolution of the DWI hyperintensities. Third, not only the ADC hypointensity but also the DWI hyperintensity persisted for at least 9 weeks. The latter is not seen in SE

The author further argues in his accompanying letter that the seizures subsided after the application of antiseizure drugs which excludes the seizures as being responsible for the MRI changes. He does not state, however, how long the seizures lasted. Since he himself calls the event "status epilepticus", prolonged seizure activity has to be assumed.

There were five witnessed focal seizures, four with generalisation, each lasting <5 min. This information was added to the case desccription.

In summary, status epilepticus as a likely cause of the MRI alterations should definitely be discussed.

We do not agree. Either the patient had a subclinical MID, or the SLL was due low thiamine, which causes mitochondrial dysfunction. A further argument against seizure activity as the cause of the cerebral lesions is their multilocality.

Reviewer 3 Report

As stated in my initial comments, sustained status epilepticus which occurred in this patient, has to be considered and discussed.
I could not see any mention of this important differential in the revised version of the manuscript.

Status epilepticus is well-known to cause similar changes as shown in this patient, and it is indeed the most likely reason for the MRI abnormalities. In particular, a pulvinar sign is often also seen as a consequence of status epilepticus.

The author argues that the persistence of the MRI changes over 9 weeks excludes status epilepticus as a possible cause. However, the drop in the ADC values reflects permanent cortical damage (cortical laminar necrosis) which can also be a consequence of status epilepticus, independent of its etiology. Thus, persistence of the MRI changes per se does not exclude SE as the cause of the MRI abnormalities.

The author further argues in his accompanying letter that the seizures subsided after the application of antiseizure drugs which excludes the seizures as being responsible for the MRI changes. He does not state, however, how long the seizures lasted. Since he himself calls the event "status epilepticus", prolonged seizure activity has to be assumed.

In summary, status epilepticus as a likely cause of the MRI alterations should definitely be discussed.

Author Response

(The authors gave the same response as above.)

Reviewer 4 Report

Thank you for the revised version of this interesting case. A few questions still remain:

1) Abstract

  • (line 18) what is the basis for the claim that the patient did benefit from B1 substitution? This is not apparent in the case description.
  • (lines 19-22) the conclusion is too strong: rather, it is possible that thiamine deficiency is associated with….AND it should be further studied whether nutritional deficiency such as especially thiamine, could give rise to secondary stroke-like lesions

2) Results

  • to be considered: addition of the facts that neuropsychological tests or genetic testing were not performed

3)  The conclusions tend to be too strong and definite for a case study

  • (line 84) first sentence, suggestion: ’possibly’ triggered
  • (lines 91-93) first passage, the last sentence, suggestion: in this case, stroke-like lesions were primarily attributed to thiamine deficiency, as the patient did not fulfill the criteria for which SLL are the hallmark...
  • line 119: references would be necessary to point out the previous descriptions of cortical lesions in WE
  • (lines 129-) suggestion: '...suggests that B1 deficiency could be associated with…., which would indicate that stroke-like lesions may occur not only in genetic mitochondrial disorders but also as a result of secondary deficits'
  • (lines 134-135): I would opt to leave out the second last sentence concerning the contribution of this case to general knowledge

Author Response

Reviewer 4

Thank you for the revised version of this interesting case. A few questions still remain:

Abstract

(line 18) what is the basis for the claim that the patient did benefit from B1 substitution? This is not apparent in the case description.

The patients’ cognitive and behavioural disturbances resolved upon administration of B1. The case description was revised accordingly

(lines 19-22) the conclusion is too strong: rather, it is possible that thiamine deficiency is associated with….AND it should be further studied whether nutritional deficiency such as especially thiamine, could give rise to secondary stroke-like lesions

Thank you for this suggestion, The conclusions were revised accordingly.

Results

to be considered: addition of the facts that neuropsychological tests or genetic testing were not performed

Was added to the limitations and case description

The conclusions tend to be too strong and definite for a case study

(line 84) first sentence, suggestion: ’possibly’ triggered

Was revised accordingly

(lines 91-93) first passage, the last sentence, suggestion: in this case, stroke-like lesions were primarily attributed to thiamine deficiency, as the patient did not fulfill the criteria for which SLL are the hallmark...

The last sentence of the first paragraph of the discussion was changed accordingly.

line 119: references would be necessary to point out the previous descriptions of cortical lesions in WE

A reference was added supporting the cortical lesion can occur in WE [Lyu et al. 2019].

(lines 129-) suggestion: '...suggests that B1 deficiency could be associated with…., which would indicate that stroke-like lesions may occur not only in genetic mitochondrial disorders but also as a result of secondary deficits'

The sentence in the conclusions was revised accordingly.

(lines 134-135): I would opt to leave out the second last sentence concerning the contribution of this case to general knowledge

The sentence was omitted.